# Nanopore electric snapshots of an RNA tertiary folding pathway

Xinyue Zhang [1,2], Dong Zhang[3], Chenhan Zhao[3], Kai Tian[1,2], Ruicheng Shi[1], Xiao Du[1], Andrew J. Burcke[1], Jing Wang[1], Shi-Jie Chen[3,4,5] & Li-Qun Gu [1,2]

The chemical properties and biological mechanisms of RNAs are determined by their tertiary structures. Exploring the tertiary structure folding processes of RNA enables us to understand and control its biological functions. Here, we report a nanopore snapshot approach combined with coarse-grained molecular dynamics simulation and master equation analysis to elucidate the folding of an RNA pseudoknot structure. In this approach, single RNA molecules captured by the nanopore can freely fold from the unstructured state without constraint and can be programmed to terminate their folding process at different intermediates. By identifying the nanopore signatures and measuring their time-dependent populations, we can "visualize" a series of kinetically important intermediates, track the kinetics of their inter-conversions, and derive the RNA pseudoknot folding pathway. This approach can potentially be developed into a single-molecule toolbox to investigate the biophysical mechanisms of RNA folding and unfolding, its interactions with ligands, and its functions.

[1] Department of Bioengineering, University of Missouri, Columbia, MO 65211, USA. [2] Dalton Cardiovascular Research Center, University of Missouri, Columbia, MO 65211, USA. [3] Department of Physics, University of Missouri, Columbia, MO 65211, USA. [4] Department of Biochemistry, University of Missouri, Columbia, MO 65211, USA. [5] Informatics Institute, University of Missouri, Columbia, MO 65211, USA. Xinyue Zhang and Dong Zhang contributed equally to this work. Correspondence and requests for materials should be addressed to L.-Q.G. (email: gul@missouri.edu) or to S.-J.C. (email: chenshi@missouri.edu)

RNAs play critical roles in the maintenance, transfer, and processing of genetic information and the catalytic control of gene expression[1]. To perform these biological functions, RNAs must fold into specific tertiary structures[2]. Thus, knowledge of the RNA folding process is key to understanding the role of RNA structures and manipulating their functionalities[3]. Various methods have been developed to explore RNA folding mechanisms, including cryo-electron microscopy[4], small-angle X-ray scattering[5], and NMR[6]. In addition to these expensive instrument-based methods, chemical probing approaches, such as selective 2′-hydroxyl acylation and primer extension (SHAPE)[7] and hydroxyl radical footprinting[8], have been widely applied to probe RNA folding structures. As RNA folding is intrinsically an intra-molecular process, various single-molecule methods, such as smFRET[9] and optical tweezers[10], have been developed to investigate RNA folding. However, due to the diffusive properties of RNA folding and short-lived transition paths, direct experimental observation of the RNA folding process, in particular the capture of intermediate folding states, remains difficult[10].

Nanopore is a promising label-free, single-molecule-based, next-generation sequencing technology[11–15]. By taking advantage of the ability to electrically detect charged biomolecules through a nanometer-wide channel, various nanopore biosensors have been developed, with targets including DNAs[16, 17], microRNA biomarkers[18–20], tRNA[21], peptides[22, 23], and proteins[24, 25] as well as epigenetic changes such as DNA methylation[26–28]. In another

important application, nanopores have been used as a precise force instrument to explore biomolecular mechanisms such as protein unfolding[29], DNA unzipping[11, 30, 31], RNA unfolding[32], and the binding of nucleic acids to enzymes[33]. Briefly, single target molecules are driven into the nanopore and produce an ion current "fingerprint" for their dissociation-translocation procedure. The nanopore under this experimental configuration is best suited to the study of unfolding-related biological problems. However, these nanopore methods thus far have not been reported to detect the other half of the biomolecular process—how a biomolecule folds into a functional tertiary structure.

In this report, we propose a generalized nanopore snapshots approach that enables elucidation of the RNA folding mechanism. This approach was established by using the gene 32 messenger RNA of bacteriophage T2[34] as a model system (Fig. 1a). This 36-nt RNA forms a typical H-type pseudoknot (PK) for regulating gene expression[35]. Our approach, as shown in Fig. 1b, allows single RNA molecules captured by the nanopore to freely fold outside the pore without constraints, starting from the single-stranded unfolded form. Remarkably, by programming the folding time, we can terminate the folding process at either the PK or any intermediate state and subsequently probe its structure based on the RNA unfolding signature in the nanopore. By characterizing the nanopore snapshots taken at different folding times and combining them with a series of theoretical analyses, we are able

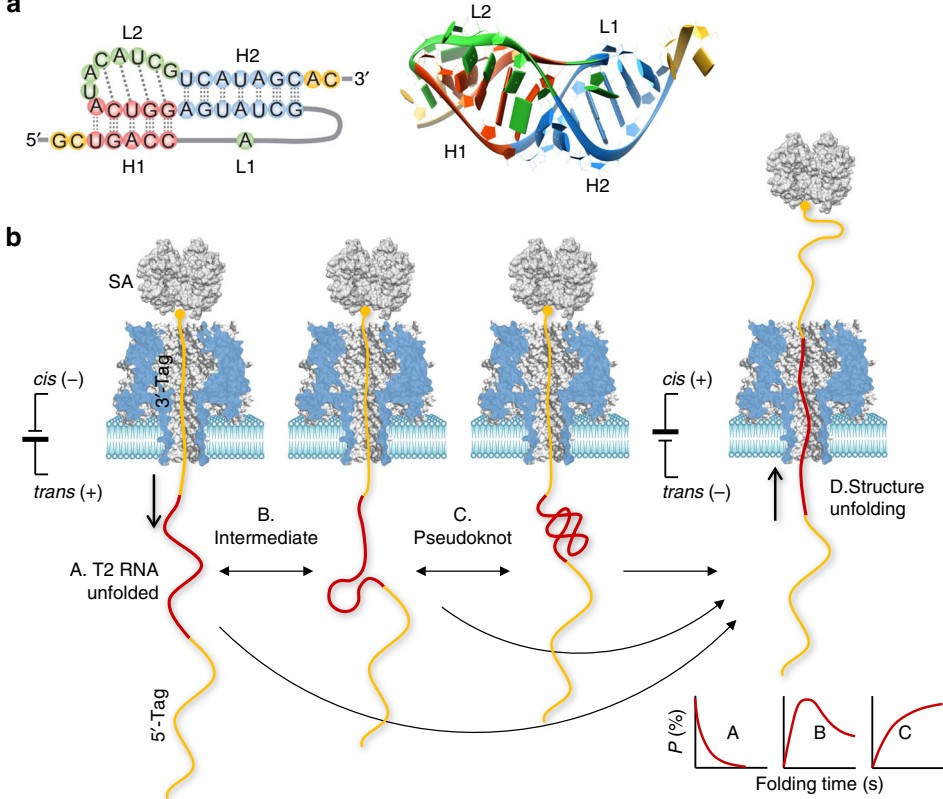

**Fig. 1** Single-molecule detection of RNA folding in a protein nanopore. **a** 2D (left) and 3D (right) structures of the pseudoknot from the 5′ end gene 32 messenger RNA of bacteriophage T2 (PDB id: 2tpk), which contains two helices, H1 (red) and H2 (blue), and two loops (green), L1 and L2. The base triple interactions between loops and helices (identified by RNAview[70]) and the coaxial stacking between the two helices are illustrated. **b** Nanopore detection of T2 RNA folding states and the folding pathway. The detection was facilitated by a complex probe containing a chimera of the 5′-DNA tag, T2 RNA, and 3′-DNA tag. The 3′ end is attached to streptavidin (SA) to immobilize the probe. Upon immobilization, the T2 RNA starts to fold from the single-stranded form in the *trans* solution. After a pre-defined folding time, the folding process is terminated at an intermediate state (B) or the pseudoknot structure (C) by using a negative voltage that disrupts the folding structure (D). The unfolding signatures reveal the folding state (B and C) and unfolding state (A). The folding time-dependent population of the identified folding states allows establishment of the pseudoknot folding pathway

**Table 1 Sequences of various complex probes used in the study**

| Probe | Sequence |
|---|---|
| T2[a] | 5′-(CAT)$_{10}$-gcugaccagcuaugaggucauacaucgucauagcac-(CAT)$_{10}$-biotin-3′ |
| T2-PolyU[b] | 5′-(CAT)$_{10}$-gcugaccagcuaugaggucauuuuuuuucauagc-(CAT)$_{10}$-biotin-3′ |
| Ref-HP[c] | 5′-(CAT)$_{10}$-gcugaccucauagcuuuuuuugcuaugagguca-(CAT)$_{10}$-biotin-3′ |

[a]T2 RNA forms a pseudoknot as shown in Fig. 1a
[b]T2-PolyU RNA is the same as T2 RNA except that loop L2 (uacaucg) is substituted with a 7-nt polyU fragment (Fig. 5a)
[c]Reference hairpin forms a 12-bp joint helix of H1 and H2, and a 7-nt polyU loop (Supplementary Fig. 2a)

to "visualize" various folding intermediates, track their inter-conversions, and derive the folding pathway.

## Results

**Designing a complex probe for RNA folding detection.** The T2 RNA PK contains two helices, H1 (5-bp) and H2 (7-bp), that are coaxially stacked on top of one another (Fig. 1a). The loop L2 (7-nt) linking the two helices interacts with H2 by non-canonical base pairing, conferring surprisingly high stability to this long quasi-continuous helix[34]. To explore the formation of the T2 PK, we designed a multi-functional complex probe, T2 (Table 1 and Fig. 1b). The probe comprises a T2-RNA extended with a poly (CAT)$_{10}$ DNA tag at both the 5′ and 3′ ends. The chimera is attached with a streptavidin at the 3′ biotinylated end, and the probe is presented in the *cis* solution of the α-hemolysin protein pore. Under a positive transmembrane voltage applied from the *trans* side (*cis* grounded), the chimera is threaded through the nanopore from the *cis* to *trans* side. As the attached streptavidin is wider than the pore entrance, it terminates chimera translocation and immobilizes the chain inside the nanopore.

Unlike previous studies that have used streptavidin-nucleic acid constructs for single-nucleotide discrimination[36, 37] and biosensor development[38–40], our complex probe was designed to possess multiple functions suited to RNA folding detection (Fig. 1b). (1) The 3′ DNA tag (30 nt, ~15 nm) is much longer than the nanopore passage (10 nm)[41]; thus, when immobilized, it can occupy the entire nanopore. This design guarantees that T2-RNA can be released into the *trans* solution, enabling RNA to freely fold without constraint from nanopore confinement. (2) Prior to folding in the *trans* solution, the T2 RNA structure is fully disrupted when passing through the nanopore. Therefore, the folding of all RNA molecules starts from the single-stranded unstructured conformation, regardless of their original folding states in the *cis* solution. (3) The folding time can be controlled such that the folding process can be terminated at an intermediate state when a negative voltage is applied to reversely pull the probe. After the pulling force unfolds the intermediate state, the generated nanopore signature can take a snapshot (i.e., report the identity) of that state. (4) As any low voltage can be applied, the nanopore can provide a minuscule pulling force (several pN) that disrupts the RNA folding structure. This behavior enables us to discover the sub-populated intermediate structures that otherwise cannot be identified under larger pulling forces. (5) In the nanopore, the folding process of the RNA is conducted at room temperature without a heating–cooling process, allowing the folding mechanism to be studied under near-physiological conditions. (6) The DNA tags of the chimera produce distinct nanopore signatures from RNA. The differences in their unique signals can be used as markers to track the RNA position in the nanopore and determine the intermediate states thereafter[32]. Note that DNA tags do not affect the RNA structure, as reported previously[32]. Similar DNA–RNA chimeras have been utilized for RNA folding detection in optical tweezers[42]. (7) As shown in Fig. 1b, the nanopore facilitated by this probe can simultaneously detect both folding and unfolding of RNA.

**Snapshots of RNA folding and intermediate states.** The overall strategy for the nanopore folding study is as follows: a positive voltage is first applied to drive a probe into the pore, disrupt its structure, and release the unstructured RNA of the probe into the *trans* solution. This unstructured molecule is held in the *trans* solution for a pre-defined duration, i.e., folding time (such as 1 s, 10 s, or 30 s) for re-folding. At the end of the folding time, a negative voltage is applied to pull the folded RNA back into the *cis* solution. Its folding structure (formed during the folding time) is then inferred from the RNA unfolding signature. Repeating this protocol enables many single-molecule snapshots to be measured, classified, and assigned to specific folding structures with a fractional population. By varying the folding time, time-dependent folding populations can be obtained to elucidate the folding pathways.

Figure 2a shows a representative current signature for the folding–unfolding of a PK in the nanopore. This experiment was performed in 1 M NaCl in the presence of 10 mM Mg$^{2+}$ (Methods). Figure 2b shows the entire molecular procedure suggested by the multi-level signature, from probe trapping to PK folding and unfolding. The probe was initially trapped in the pore at +120 mV. As the probe passed through the pore, its different domains sequentially occupied the pore lumen, resulting in the stepwise change in the block level (conductance). When the 5′ DNA tag was first threaded in the pore, the current was reduced to Level-1 ($I/I_0 = 18.6 \pm 1.6\%$, A). Next to the 5′-tag, the T2 RNA was pulled into the pore while disrupting its initial structure, further reducing the current to Level-2 ($12.6 \pm 1.5\%$, B). Consistent with the previous finding, the RNA in the nanopore reduced more conductance than DNA in the pore[19, 32]. As the 3′ DNA tag entered and was immobilized (by the attached streptavidin) in the pore in place of T2 RNA, the nanopore current was returned to Level-1 (C). At this moment, the T2 RNA that was released into the *trans* solution started to re-fold (D). The folding time ($t_{fold}$) for this illustrated event was 10 s. By the end of $t_{fold}$, the voltage switched to −60 mV to pull the probe backward (from *trans* to *cis*) to unfold the RNA structure. This process produced the characteristic two-level block pattern, a long block at Level-3 ($I/I_0 = 9.2 \pm 0.5\%$, E), followed by a short deeper block at Level-4 ($I/I_0 = 3.0 \pm 0.7\%$, F). From Level-4, the current returned to the open pore level (G). According to a previous study[32], this type of block pattern is the signature for the stepwise unfolding of a PK. The total duration of Level-3 and Level-4 was the unfolding duration ($\tau$) of PK. Here, $\tau = 3500$ ms $\pm 330$ ms at −60 mV (Fig. 3a) and can be dramatically shortened to 39 ms $\pm 6$ ms at −120 mV (Supplementary Fig. 1). To interpret the high stability of the PK, we constructed a reference hairpin ref-HP (Table 1 and Supplementary Fig. 2). Ref-HP utilized all the base pairs of H1 (5 bp) and H2 (7 bp) of the T2 PK to form a long helix (12 bp) but without loop–helix interactions. $\tau$ for Ref-HP unfolding was only 18 ms $\pm 3$ ms at −60 mV (Supplementary

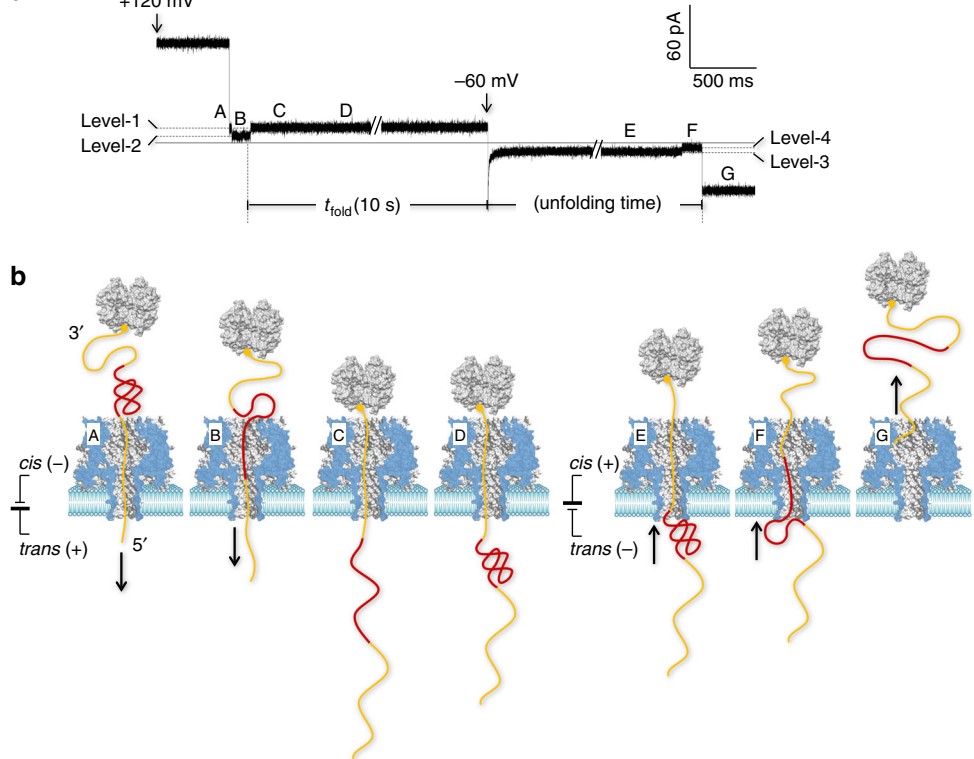

**Fig. 2** Nanopore snapshot for T2 RNA pseudoknot formation. **a** Stepwise multi-level nanopore current signature generated by a complex probe under the +120 mV/$t_{fold}$ = 10 s/−60 mV protocol (the probe was trapped at +120 mV, the folding time $t_{fold}$ was 10 s, and the folding structure was unfolded at −60 mV). **b** Model cartoons showing the entire trapping–folding–unfolding procedure revealed by the signature in **a**: translocation of the 5′ DNA tag (A, Level-1), disruption while translocating T2 RNA (B, Level-2), translocation and immobilization of the 3′ DNA tag (C, Level-1), folding of the pseudoknot from the single-stranded RNA on the *trans* side (D) for a pre-defined folding time ($t_{fold}$ = 10 s), unfolding of the folding structure under back-pulling at −60 mV while the 3′ DNA tag remains in the pore (E, Level-3), translocation of the partially unfolded T2 RNA through the pore (F, Level-4), and a fully disrupted probe pulled out of the pore and back to the *cis* solution (G). The time from the beginning of negative voltage (−60 mV) application to the block end is the unfolding duration of an RNA folding state ($\tau$), during which the folding structure was disrupted

Fig. 2), consistent with the reported results for hairpin[40]. Therefore, Ref-HP is 200-fold less stable than PK, verifying the significant contribution of the loop–helix interaction to the stability of the PK[32]. Overall, through a nanopore folding–unfolding electric signature, we can capture a folding snapshot for the formation of a PK from single-stranded T2 RNA in a given folding time.

Figure 3a shows the unfolding duration distribution of the folding snapshots with 10 s folding time. We identified the PK state (3500 ms) as well as three less stable components at 2.5 ms ± 2 ms, 20 ms ± 3 ms, and 210 ms ± 25 ms. Each identified component corresponds to a non-PK intermediate state. Figure 3b shows the representative current traces for each type of folding state. Note that the short 2.5 ms and 20 ms states could not be seen at high voltage, such as −120 mV (Supplementary Fig. 1), supporting the notion that the unique small pulling force provided by the nanopore (~5 pN at −60 mV[32, 43, 44]) is the key to discovering less stable intermediates. In addition to the various folding states identified above, we occasionally observed an open pore current when negative voltage was applied (Fig. 3b). These events were generated by the unfolded T2 RNA in the single-stranded (SS) form that was rapidly pulled back to the *cis* solution.

To provide insight into the structural details of intermediate folding states, we conducted coarse-grained (CG) molecular dynamics (MD) simulation. For the T2 RNA sequence, the simulation revealed that three intermediate structures (Fig. 3c)

emerge before the final formation of the native PK. The initial coil state, SS, undergoes transitions to two possible stem–loop structures, HP1 and HP2. HP1 is a hairpin that contains the native 7-bp helix H2 (the longer stem in the T2 PK), with three additional non-native base pairings (bps) formed in the loop region. HP2 is a fully non-native stem–loop structure consisting of 11 canonical bps along with two or more non-canonical bps. The simulation also revealed a non-native PK-like intermediate, the "TS" state. The TS structure contains the native stem H2 and a misfolded stem P′. A distinct feature of TS is the non-native tertiary interaction between stem P′ and the loop. The misfolded HP2 and TS structures are also supported by separate computational models. For example, the Vfold-based free energy analysis for the 2D structures[45] and the SimRNA-based 3D structure analysis[46] predicted TS and HP2 as misfolded suboptimal states of the sequence.

Overall, the observations revealed by the CG MD simulation are in agreement with the above nanopore experimental results. The nanopore folding snapshots and the CG MD simulation can be combined to construct the structural details for the observed intermediated states. In other words, by considering the features of unfolding nanopore signatures and the stabilities of the intermediated states relative to the ref-HP, we can establish connections between the intermediate states identified by the nanopore folding snapshots and the kinetics intermediates observed in the CG MD simulation. Compared with 12-bp ref-HP (18 ms ± 3 ms at −60 mV), 7-bp HP1 should be less stable

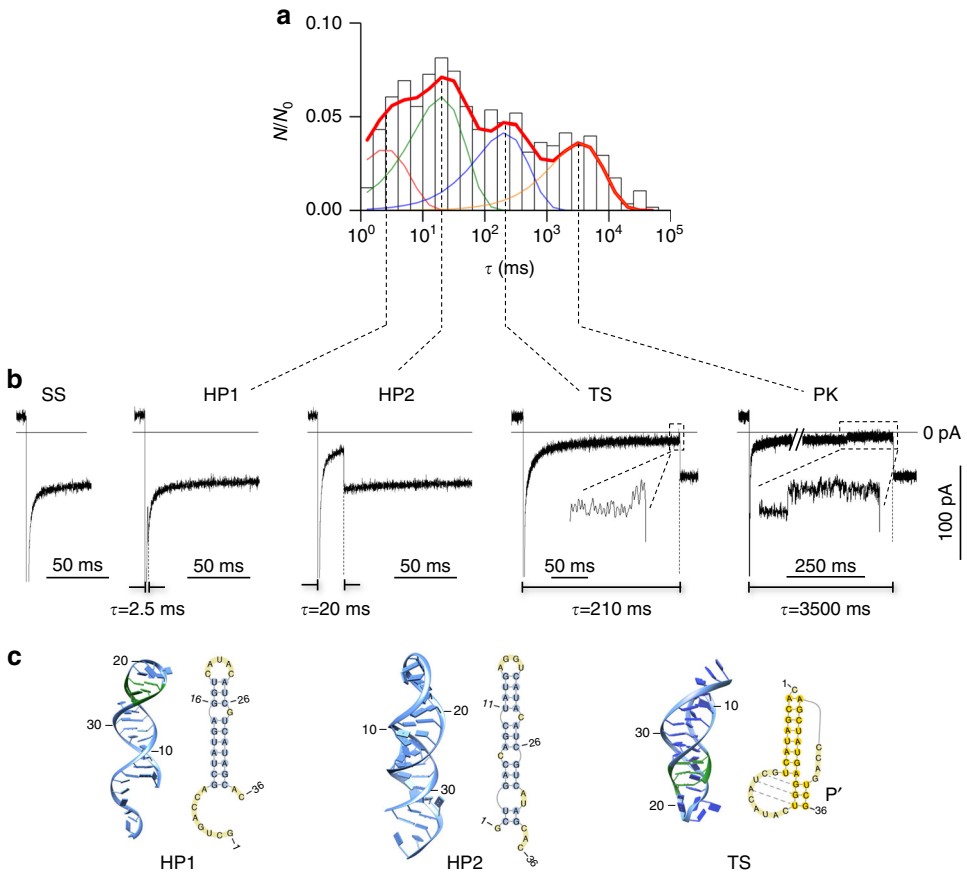

**Fig. 3** Detection of intermediate folding states. **a** Histogram showing the distribution of unfolding durations of various folding states ($N_0 = 578$). The x-axis is the time with bin width in the log scale. The y-axis is $N_0/N$, where $N_0$ is the total number of events in the histogram and $N$ is the number of events within each bin (log time). $N/N_0$ represents the fractional population for each bin, and the sum of $N/N_0$ for all bins, i.e., the area of the histogram, is normalized to "1". After fitting, the area covered by each component is the fractional population of the corresponding structure (Fig. 4). In this histogram, the unfolding duration of each RNA event was determined under the +120 mV/$t_{fold}$ = 10 s/−60 mV protocol. Four components at 2.5 ms, 20 ms, 210 ms, and 3500 ms were identified in the distribution (Methods). Each component corresponds to a folding state. **b** Representative nanopore current signatures for the four folding states identified from the unfolding duration distribution in panel a. The longest component (3500 ms) is for the pseudoknot (PK). The 2.5 ms, 20 ms, and 210 ms components are intermediate states. The two insets clearly show the two-level conductance patterns, TS and PK, which are distinct from the single-level patterns for the other intermediate states, HP1 and HP2. In addition, the signature of the open pore current (without block) for single-stranded RNA (SS) is shown. **c** Structures observed in the simulation that correspond to the three intermediate folding states identified in the nanopore experiment, HP1 (2.5 ms), HP2 (20 ms), and TS (210 ms). 2D and 3D structures for the three intermediates were found in the CG MD simulation. Three non-native base pairs in HP1 and the misfolded stem P′ in TS are labeled in green

and thus is assigned to the 2.5 ms state. The 13-bp HP2 corresponds to the 20 ms state due to it similar stability to ref-HP. In addition, the single-level blocks (−60 mV) for the two states (HP1 and HP2 in Fig. 3b) are consistent with the hairpin structure that is unfolded in one step (Supplementary Fig. 2). TS can be assigned to the 210 ms component because it is much more stable than ref-HP, consistent with the stabilization effect of the loop–helix interactions observed in TS. Moreover, the unfolding of TS produced a two-level block pattern (−60 mV, Fig. 3b), consistent with its PK-like structure obtained in the simulation. In summary, based on the nanopore snapshot and CG MD simulation, we identified a series of T2 RNA folding states and their structures, including single-stranded SS, intermediates HP1, HP2 and TS, and native PK.

**RNA folding pathway**. After identifying all the folding states, we combined the nanopore snapshot data and theoretical analysis to investigate the time-dependent folding process and the transitions between the different states for the purpose of establishing a PK

folding pathway. Experimentally, we investigated how the folding of T2 RNA varied with the folding time. We first obtained a series of nanopore folding snapshots, which are characterized by the unfolding duration distributions, with different folding times ranging from 1 s to 60 s (Supplementary Fig. 3). We then calculated the fractional populations of all the identified states (SS, HP1, HP2, TS, and PK) in the unfolding duration distributions (Supplementary Fig. 3). Finally, we obtained the population kinetics (P–$t_{fold}$ curve) for each state (Fig. 4a and Methods). We found that the population of the SS state sharply decreased with increasing folding time. For HP1, the population increased very quickly in the initial stage, followed by a rapid decrease in the population. For HP2 and TS, the population increased before folding time $t_{fold}$ = 10 s and then slowly decreased. In contrast, the population for the native PK state of T2 RNA continuously increased before finally reaching an equilibrium with $t_{fold}$ = 60 s. All states reached equilibrium around the folding time of 60 s.

To understand these time-dependent folding populations, a folding model based on the master equation approach[36, 37] was developed (Fig. 4b). Fitting the experimentally determined

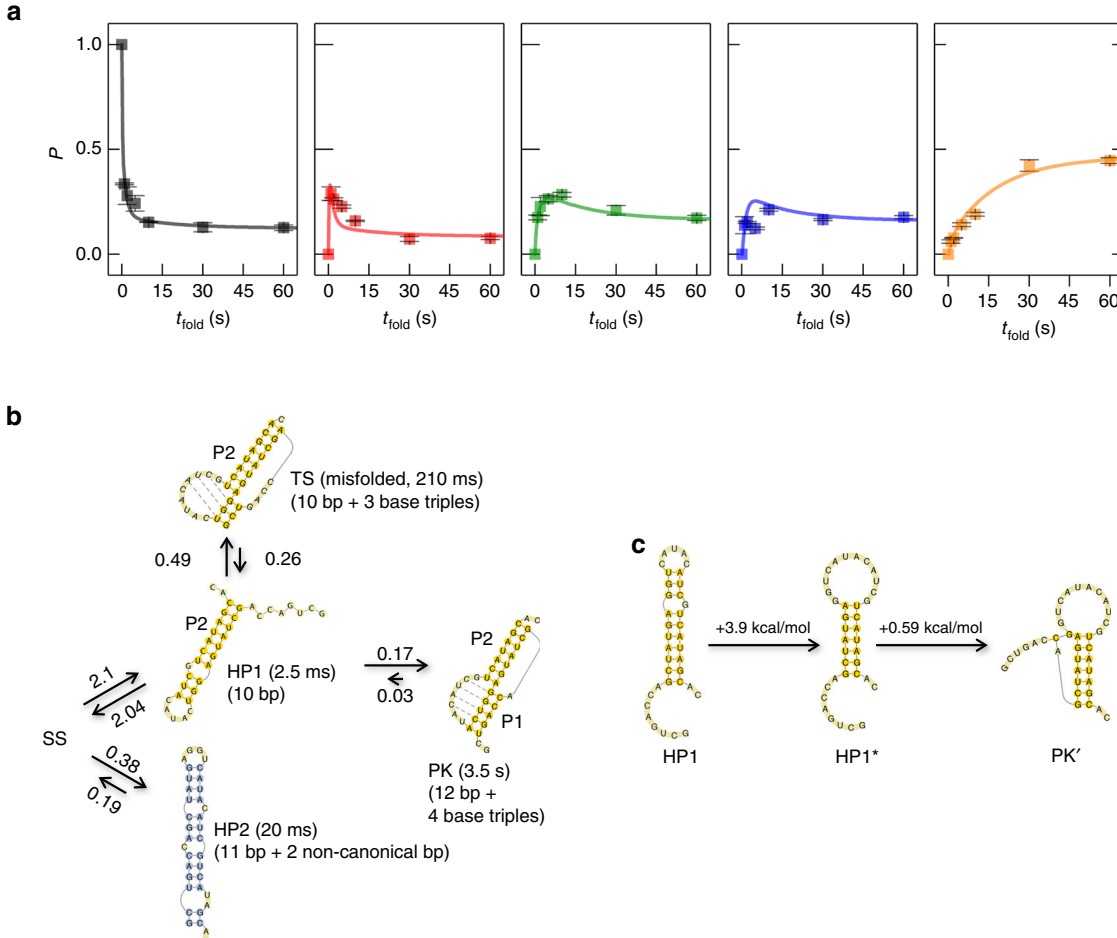

**Fig. 4** Folding pathway of the T2 RNA pseudoknot. **a** The fractional population of the different states (from left to right, SS, HP1, HP2, TS, and PK, respectively) as a function of the folding time. The solid squares represent the experimental data. Error bars represent standard deviation of the mean ($n \geq$ 3). We first obtained the unfolding duration histogram at different folding times (Supplementary Fig. 3). Each histogram was then fitted globally with a sum of four exponentials, and the normalized area of each component is the fractional population of the corresponding state (Methods). The curves in all plots of the panel were theoretically predicted based on the master equation approach. **b** The predicted pseudoknot folding pathway based on the analysis of the time-dependent population distribution and from the simulation. In the folding pathway of T2 PK, starting from the single-stranded (SS) state, the RNA can form two hairpin structures, HP1 and HP2. From HP2, the RNA can only go back to the SS state. Disruption of the non-native base pairs and loop–helix base triples in the misfolded TS state results in the HP1 structure. Therefore, the TS state can lead to the native PK structure via HP1 as the intermediate. Also shown in the figure are the fitted rate constants (in s$^{-1}$) for the transitions between the states and the unfolding durations (shown in the parentheses) of each state under the +120 mV/$t_{fold}$ = 10 s/−60 mV protocol. **c** Two "uphill" steps in the HP1→PK folding process: disruption of the three non-native base pairs in the HP1 hairpin and the loop formation and the subsequent initiation of helix H2 formation in the native PK structure

population kinetics for the five states (Fig. 4a) yielded the estimated rate constants (Fig. 4b and Methods). SS is the initial state of the pathway for all the folding states observed. The RNA chain in the SS state quickly folds into the on-pathway intermediate HP1 and the off-pathway intermediate HP2. The initial folding from SS results in a quick decrease in the SS population and a concurrent initial increase in the HP1 and HP2 populations (Fig. 4a). In this initial process, approximately $k_{SS \to HP1}/(k_{SS \to HP1} + k_{SS \to HP2}) = 84.7\%$ and $k_{SS \to HP2}/(k_{SS \to HP1} + k_{SS \to HP2}) = 15.3\%$ of the population goes from SS to the HP1 and HP2 states, respectively. We note that HP1, TS, and PK share the same long native helix H2 (Fig. 3c), and HP2 is fully non-native and adopts a completely different fold from HP1, TS, and PK. Therefore, apart from going back to the SS state, direct transitions between HP2 and HP1, TS, or PK can be ignored. As a result, the HP2 population experiences a slow decrease and then reaches a plateau (Fig. 4a). From HP1, the RNA can fold directly to either the native PK or the misfolded TS state or back to the initial SS state. After the formation of the HP1 state, 6.3% of the

initial HP1 population would fold directly to the native PK, ~18.1% to the misfolded TS state, and 75.6% back to the SS state. These transitions together cause the decrease of the HP1 population and the increase of the PK and TS populations (Fig. 4a). The TS structure cannot directly fold to PK. Instead, TS returns to HP1 via the disruption of the misfolded base pairs. Therefore, a slow decrease in the TS population occurs as a result of the detrapping transition from TS to HP1, which subsequently folds to the native PK state (Fig. 4b). In addition, based on the Vfold RNA folding model[47, 48], the HP1 → PK transition is rate limited by the (mainly enthalpic) barrier (~ 3.8 kcal/mol) to break the three non-native intra-loop base pairs in the HP1 hairpin loop in the HP1 → HP1$^*$ transition and the (mainly entropic) barrier (~0.59 kcal/mol) for the formation of the helix stem H1 in the HP1$^*$ → PK′ transition (Fig. 4c). As the former step has a higher barrier, it may be the rate-limiting step for the HP1 → PK folding process.

In summary, the T2 PK folding process follows two possible routes starting from SS (Fig. 4b). In route I, SS first forms a short

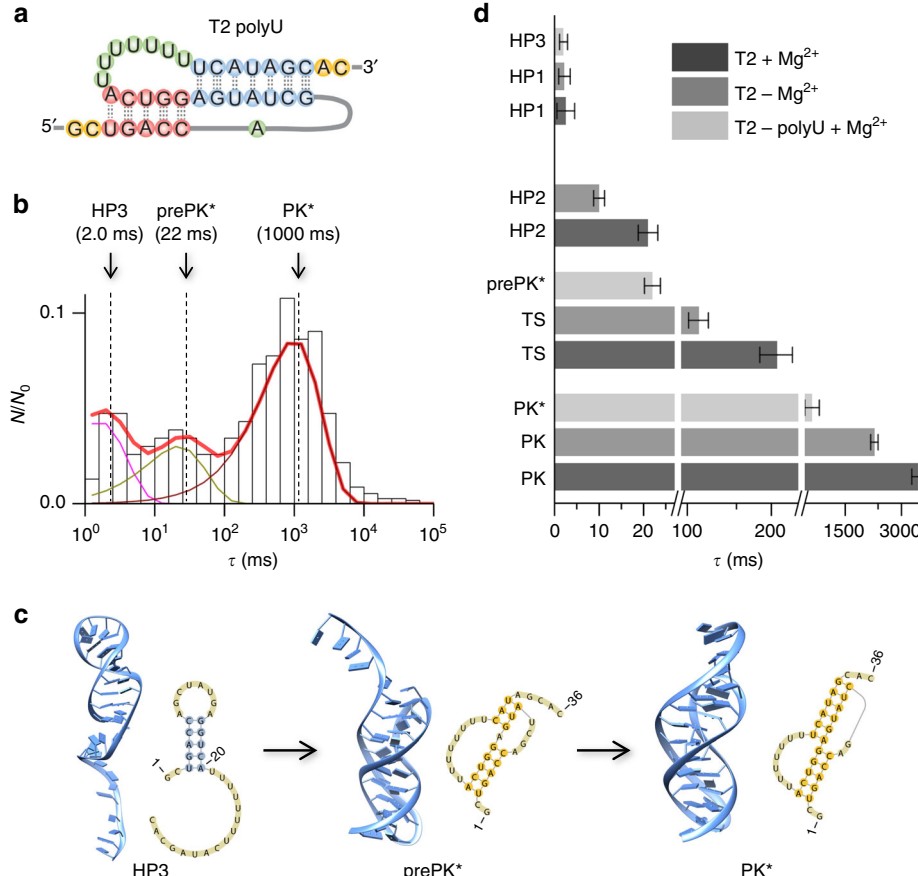

**Fig. 5** Regulation of the pseudoknot folding pathway by RNA sequence and magnesium ions. **a** 2D structure of T2 RNA with a polyU loop. **b** Histogram of the unfolding duration for T2 polyU mutant folding states after the folding time of $t_{fold} = 10$ s ($N_O = 232$). Three components at 2.0 ms for HP3, 22 ms for prePK*, and 1000 ms for PK* were identified from the histogram. **c** 2D and 3D structures for the intermediate states HP3 and prePK* and the final state PK* obtained from the simulation study. **d** Comparison of unfolding durations of various states in wild-type T2 with magnesium ions, wild-type T2 without magnesium and T2 with polyU loop with magnesium ions. Error bars represent standard deviation of the mean duration ($n \geq 3$). Magnesium ions are at the concentration of 10 mM, and the unfolding voltage in *trans* is −60 mV. All the data were collected from at least three different nanopores, and the errors are given as standard deviations

hairpin HP1. This hairpin contains a 7-bp native helix H2 in the T2 PK and three non-native base pairs inside the loop. The HP1 → PK folding involves two parallel pathways: direct folding to PK and folding through the misfolded TS state. This structure can reversibly return to HP1 and then folds to PK. In Route II, SS folds to a 13-bp hairpin HP2 with three bulges. This structure is stable, and unfolding back to the single-stranded SS state is slow. Route I and Route II together form the folding pathway of T2 RNA.

**Regulation of the folding pathway by loop and magnesium ion.** Since the loop–stem interactions in the PK are critical for structural stability[3, 49], we investigated how these interactions influence the folding pathway. To weaken the loop–helix tertiary interactions, we changed the sequence of loop L2 into polyU (Fig. 5a, Table 1). Nanopore experimental data showed that this T2-polyU mutant has only three folding states (Fig. 5b). The first state had an unfolding duration of 2.0 ms ± 1.1 ms, which is similar to that of the 7-bp hairpin HP1 (2.5 ms) formed in Route I of the wild-type PK folding. In combination with the CG MD simulation results, we assigned this state to a short hairpin (HP3, Fig. 5b, c) that contains the 5-bp native helix H1 of the native PK. The second intermediate state had an unfolding duration of 22 ms ± 3 ms (Fig. 5b), and the simulation showed that this state may be a PK-like structure, with a native helix H1 and partially

folded helix H2 (6–8 bp in total) and without a loop–helix interaction (prePK*, Fig. 5b, c). Compared with the wild-type PK, the helix stem H2 was not fully stabilized because the polyU mutant lacks the loop–helix base triple tertiary interactions in the wild-type PK. Although the polyU loop could significantly disrupt the loop–stem interaction, it may form weak loop–stem interactions to accommodate the folding to the final state of this sequence (PK*), which had an unfolding duration of 1000 ms ± 80 ms in the nanopore (Fig. 5b). This interpretation agrees with our previous finding for the PK unfolding kinetics[32]. Overall, the loop–stem tertiary interactions are important for the PK and its folding process.

Magnesium ions ($Mg^{2+}$) can regulate RNA structure formation and stabilize the pocket motifs in RNA or DNA tertiary structures[50–52]. To explore the effect of $Mg^{2+}$ on the PK folding process, we repeated the nanopore snapshots in 1 M NaCl in the absence of $Mg^{2+}$ (Methods). With 10 s folding time, T2 RNA without $Mg^{2+}$ folded to four states, similar to T2 with $Mg^{2+}$ (Fig. 5d). The unfolding durations of the HP1 and HP2 states (at −60 mV) were similar to those with $Mg^{2+}$ ions, indicating that the two hairpin-based states are less influenced by $Mg^{2+}$. In contrast, without $Mg^{2+}$, the two PK-like states, TS and PK, were less stable, with the unfolding durations shortened from 3500 ms ± 330 ms to 1900 ms ± 170 ms for PK and from 210 ms ± 25 ms to 80 ms ± 9 ms for TS, respectively (Fig. 5d). This change might have been

caused by a reduction in ion binding and charge neutralization as well as the loss of interactions between $Mg^{2+}$ and the PK loops. The $Mg^{2+}$–loop interaction stabilizes the PK structure but not the first two less stable states (hairpins HP1 and HP2), which lack loop–helix interactions and have much weaker stabilizing interactions with $Mg^{2+}$.

## Discussion

The nanopore snapshot-based kinetic detection, combined with CG molecular dynamics-based structural modeling, master equation-based rate constant estimation, and population kinetics analysis, provides a tool to investigate the RNA folding process. Through a programmable RNA disruption–refolding–disruption procedure in the nanopore, a series of intermediates that may be inaccessible in equilibrium experiments can be captured based on their unfolding signatures and the population kinetics for each state can be measured. The nanopore is able to capture a wide range of intermediates, with the unfolding duration ranging from milliseconds to seconds and minutes. The CG MD simulations can identify the potential intermediate structures and kinetic pathways, which are supported by the nanopore signatures. Based on the identified structures, the master equation approach can provide the transition rates between different states from the population kinetics. The combination of the above three methods leads to a reliable construction of the folding kinetics. The reliability of the results is supported by the theory–experiment consistency for the different measured properties. The integrated approach above can potentially be adapted for the study of RNAs with unknown structures. For RNAs with unknown structures, in addition to the structure information provided from the computational studies, nanopore data for various designed mutants would be highly useful for the probing and confirmation of the structures, stabilities, and folding pathways. The theory–experiment comparisons for the designed mutants may lead to reliable identification of the key factors that determine the kinetics.

Our study suggests five-state folding kinetics for the T2 RNA PK. Similar to the unfolding kinetics for the same molecule revealed in a previous study[53], the folding kinetics are multi-state and non-cooperative and involve off-pathway misfolded intermediates and the kinetic pathways are strongly influenced by the helix–loop tertiary interactions. In addition, the comparative studies for the wild-type T2 PK and the polyU mutant indicate that the sequence forming the same PK may fold through very different pathways[54]. Among the three folding intermediates identified, two are hairpins, and the third is a PK-like structure that adopts a short misfolded stem for one of two helices in the native structure. The rate-limiting step of the whole folding pathway is the formation of the native PK from the hairpin structure that contains a native helix and three extra base pairs in the hairpin loop region. The folding equilibrium within an RNA PK is highly sequence dependent (mutation of one loop with polyU can change the stability of the PK) and can be regulated by magnesium ions. These results have important implications for the interplay between RNA structures and their functions in cells, since the regulatory functions of RNA molecules are often related to conformational transitions[55].

To identify more intermediate states (such as the pre-PK structure with two native helices formed but without loop–helix tertiary interactions[53, 54]) and to develop further applications, the resolution and accuracy of the nanopore system need improvement. First, due to the resistor–capacitor (RC) time of the setup, the voltage change will simultaneously cause a curved charging current (Figs. 2a and 3b). Rapidly unfolding events occurring in this sharply changing current region may be unidentifiable. To improve the temporal resolution for the detection of fast events, the RC time of the setup requires optimization. Meanwhile, the application of low voltage can significantly prolong the unfolding duration for identifying fast, partially folded structures. Second, it could be difficult to discriminate different folding structures that are similar in unfolding duration (Fig. 3a) due to their large population overlap in the duration histogram. One solution is to use the blocking level as an "identifier" to discriminate different folding structures. As verified in Supplementary Fig. 4, different lengths and positions of RNA in a DNA–RNA chimera can be clearly discriminated based on their characteristic blocking levels in the nanopore, providing the potential to use the blocking level for precisely identifying RNA folding states. Third, how to apply this approach for long RNA investigations remains an issue. We anticipate that combining it with alternative methods, such as nanopore sequencing[14], may help to precisely read RNA positions and report various folding states. Notably, synthetic nanopores with tunable dimensions have been developed recently to characterize drug-induced RNA conformational changes[56]. Combined with machine learning, synthetic nanopores can discriminate different tRNAs[57].

From the theoretical aspect, a more physically reliable force field in the MD simulation is helpful to provide structural details for various potential folding intermediate states and to understand the nanopore unfolding current signature. The key points include non-canonical base pair interaction, multiple base–base interaction, nucleotide-ion interaction, freedom from native bias, the ability to account for the whole folding landscape, and more efficient conformational space sampling. Furthermore, a direct simulation of the nanopore unfolding and translocation experiment using methods such as steered molecular dynamics could be helpful to understand the unfolding electric signatures of various structures and to establish a direct connection between the nanopore signature and RNA structure. These efforts would significantly improve the resolution of the nanopore technique, making it possible to more accurately reveal RNA intermediate states and folding pathways.

The improved system has the potential to study a variety of disease-relevant RNA and DNA tertiary structures, from hairpins[58] and PKs[59] to kissing-loops[60] and G-quadruplexes[61], and RNA-based interactions, from microRNA–target RNA interactions to ligand–RNA interactions, such as RNA repeats[62] and riboswitches[63]. In addition to mutant construction, modified nucleotides can be introduced to detect chemical-specific folding procedures in the nanopore. Therefore, the nanopore can be potentially combined with chemical approaches, such as selective 2′-hydroxyl acylation analyzed by primer extension (SHAPE)[64], for joint structure exploration. Most notably, our nanopore system can provide very small forces (by applying low voltage) to detect weak interactions in small RNA structures. As demonstrated by this and previous studies[32], this system is also able to characterize non-canonical base pairs involved in tertiary structure formation and dissociation. Finally, this system may be generalized to investigate the folding of other biomolecules, including polymers, peptides, and even proteins. Overall, this method can find applications in biomolecular folding investigations and related areas such as pharmaceutical kinetic investigations and drug development.

## Methods

**Materials and probe formation**. All chemicals, including NaCl, $MgCl_2$, 3-(N-morpholino) propanesulfonic acid (MOPS), and diethylpyrocarbonate (DEPC), were purchased from Sigma-Aldrich (St. Louis, MO, USA) and used as received. Lipid 1,2-diphytanoyl-sn-glycero-3-phosphocholine for bilayer formation was purchased from Avanti Polar Lipids (Alabaster, AL, USA) and used without further purification. All biotinylated RNA–DNA chimeras (Table 1) were synthesized and purified by Integrated DNA Technologies Inc. (Coralville, IA, USA), and dissolved

in Millipore water to a stock concentration of 100 μM. Streptavidin was purchased from ProSpec-Tany Technogene Ltd. (East Brunswick, NJ, USA). The recording solution contained 1 M NaCl, 25 mM MOPS, and 10 mM MgCl₂ (pH 7.4). In the magnesium effect experiment, different concentrations of MgCl₂ from 0 to 10 mM were used in the recording solution. In total, 5 μL of the chimera stock solution was directly released to the *cis* solution to reach a final concentration of 250 nM. Then, 25 μL of 100 μM streptavidin solution (in 100 mM NaCl, 25 mM MOPS, pH 7.4) was added to the *cis* compartment. The mixture was incubated for 15 min in order for streptavidin to bind the chimera and form the complex.

**Nanopore formation and measurement**. The nanopore recording chamber was assembled by two Teflon compartments that were separated by a thin Teflon partition film (Goodfellow Corp., Coraopolis, PA). Each compartment was filled with the recording solution to support the lipid bilayer formation and facilitate the ionic current flow. A lipid bilayer membrane was formed spanning a 150 μm orifice fabricated in the center of the partition. About 1 μL of the α-hemolysin protein solution was released into the *cis* compartment. The protein can be spontaneously inserted into the lipid membrane to assemble a single nanopore. The ionic current through the nanopore at various transmembrane voltages was monitored by an Axopatch 200B amplifier (Molecular Devices Inc., Sunnyvale, CA, USA) and acquired by a DigiData 1440A A/D converter (Molecular Devices), filtered with a built-in 4-pole low-pass Bessel filter at 5 kHz with a sampling rate of 20 kHz. A Clampex software (Molecular Devices) was used for the data recording and acquisition, and a Clampfit software (Molecular Devices) was used for analyzing the nanopore current traces, including event duration histogram analysis and amplitude histogram analysis.

**Extraction of folding state lifetimes**. The folding of RNA to a specific tertiary state in the pathway is a time-dependent procedure. Tracking the time-dependent formation of each state in the nanopore allows the establishment of the RNA folding pathway. This process is realized by stopping the folding procedure after a given folding time ($t_{fold}$), and each folding state can occur with a specific probability ($P$). The folding procedure can be stopped at an intermediate structure or the native PK structure with a probability ($P$) by pulling the probe reversely in the *trans*-to-*cis* direction with a negative voltage (–60 mV). Under this pulling force, the formed folding structure can be disrupted. The duration from the beginning of voltage application to the block end for structure disruption is measured as the lifetime of the structure ($\tau$). Each folding structure has a specific stability and can be identified from its lifetime. The lifetime of a folding structure follows the exponential distribution. According to previous studies[65, 66], if the bin is log($t$), the distribution can be expressed as

$$f(t) = c\mathrm{e}^{(\ln t - \ln \tau) - \mathrm{e}^{(\ln t - \ln \tau)}}, \qquad (M1)$$

where the amplitude $c$ is equivalent to the area covered by the distribution. For multiple states, the lifetime distribution can be expressed as

$$f(t) = \sum_{i=1}^{n} \left[ c_i \mathrm{e}^{(\ln t - \ln \tau_i) - \mathrm{e}^{(\ln t - \ln \tau_i)}} \right]. \qquad (M2)$$

The fractional population of the state $i$, $P_i$, can be obtained from

$$P_i = (1 - P_0) c_i / \sum_{i=1}^{n} c_i, \qquad (M3)$$

where $P_0$ is the population of the unfolded single-stranded state (SS), which is the beginning state of all RNA molecules measured, and $1 - P_0$ is the total fraction of RNA molecules that form various folding structures (including intermediate and PK structures). $P_0$ is obtained by measuring the number of SS events ($N_0$, events without blocks) divided by the total number of events collected.

We first constructed the lifetime distributions (histograms) for RNAs experiencing a folding time of 1 s, 2 s, 5 s, 10 s, 30 s, and 60 s and then fit each histogram with four components (HP1, HP2, TS, and KP) by using Eq. (M2). Finally, the fractional populations were obtained using Eq. (M3). For $\tau$ and fractional population, the mean and standard deviation (SD) were obtained from at least three experiments ($n \geq 3$) with independent nanopores. All nanopore pulling experiments were conducted at room temperature (22 ± 1 °C).

**Molecular dynamics simulation method**. Based on a CG RNA model, where the pyrimidines/purines are represented by 4/5 CG beads and a knowledge-based force field, we simulated the folding of the wild-type T2 and T2-polyU mutation RNAs using Langevin dynamics with the modified LAMMPS packages[67]. To enhance the conformational sampling, we used Replica-Exchange MD (REMD) with 10 replicas for temperatures ranging from 175 K to 400 K. In the simulation, the total simulation time per replica was set to $t = 1$ μs, with an integration time-step $\Delta t = 0.5$ fs. To monitor the folding process, the conformational snapshots were collected every 50 ps. For every 25 ns simulation time interval, the collected snapshots (5000 structures in total) were submitted to the clustering procedure. Based on the pairwise root mean square deviation (RMSD) for all the CG beads within a cutoff of 5.0 Å, the top 50% low-energy structures (2500 structures for every simulation

interval) were clustered to identify the typical intermediate states. The sequence of centroid structures of the clusters gave the folding pathway.

**Master equation method**. Based on the initial unfolded state, the native PK state, and the intermediates states, we constructed the master equation[68,69] to estimate the transition rates between the different states. For a system of $\omega$ states, the master equation method considers the rate of population changes ($\mathrm{d}P^i/\mathrm{d}t$) for the $i$th state as the difference between the rates of entering and leaving the state:

$$\frac{\mathrm{d}P^i}{\mathrm{d}t} = \sum_{j \neq i} (k_{ij} p^j - k_{ji} P^i), \qquad (M4)$$

where $P^i$ is the population of the $i$th state and $k_{ij}$ and $k_{ji}$ are the rate constants for the transitions from states $j$ to $i$ and from $i$ to $j$, respectively. With a column vector $\vec{P} = \mathrm{col}\,(P^1, ..., P^\omega)$ to represent the population of each state, the master equation can be transformed into a matrix form $\mathrm{d}\vec{P}/\mathrm{d}t = M \cdot \vec{P}$. In that formula, **M** is a $\omega \times \omega$ rate matrix. Off-diagonal elements inside the matrix are defined as $M_{ij} = k_{ij}$, and the diagonal elements are $M_{ii} = -\sum_{j \neq i} k_{ji}$.

To obtain the time-dependent population of each state $\vec{P}(t)$, we found the eigenvalues $\lambda_\mu (\mu = 1,2, ..., \omega)$ and eigenvectors $\overrightarrow{n_\mu} (\mu = 1,2, ..., \omega)$ of rate matrix **M**. The time-dependent population of the different states can be calculated as $\vec{P}(t) = \sum_{\mu=1}^{\omega} c_\mu \overrightarrow{n_\mu} \mathrm{e}^{\lambda_\mu t}$, where the coefficients $c_\mu$ are determined from the initial population of each state.

For the T2 PK folding, our simulations suggested three intermediate states in addition to the initial unfolded and final folding states (see main text for details). For such a five-state system, the master equation involves a 5×5 rate matrix. By solving the master equation, we estimated the rate constants from the best fitted, experiment-determined population kinetics.

**Parameter optimization**. The fitness function $F$ is defined as the RMSD between the experimentally determined population and the theoretically predicted population for each state:

$$F = \sqrt{\frac{\sum_i \sum_t \left( P_E^{(i)}(t) - P_T^{(i)}(t) \right)^2}{N}}, \qquad (M5)$$

where $P_E^{(i)}(t)$ and $P_T^{(i)}(t)$ are the experimentally measured and the theoretically predicted populations for state $i$ at time $t$, respectively, and $N=30$ is the total count of experiment data (i.e., the number of time points sampled in the sum over time $t$ in the fitness function).

One of the states in the master equation calculation is the unfolded state, denoted as SS (single-stranded state). In the experiment, the SS state may be a fraction of the total ensemble of "unfolded" conformations identified from the electric current. In the experiment, the population of the SS state is estimated from the events where the electric current is not blocked. Limited by the experimental conditions, such as the purity of the RNA sequence and the effect of DNA extension, it is possible that not all the SS RNA chains can be involved in the folding process. There exists a buffer state in the experiment, $SS_0$, which should be deducted from the apparent total SS population in order to calculate the effective population of SS. Therefore, in the theory–experiment fitting process, the effective SS population is equal to the apparent (experimental data-derived) total SS population minus the population of the buffer state $SS_0$. Our estimation based on the experimental data for the populations of the different states (see main text for the details) suggested that [$SS_0$] is ~4.2% of the total initial SS population. To avoid the possible trapping in local minima in the optimization procedure for the fitness function $F$, the parameter search process was repeated with different initial values. Additionally, to further confirm the fitted rate constants, we used alternative algorithms, such as the genetic algorithm, for the optimization. Comparisons between the different methods also led to the same consistent results.

**Data availability**. All data related to this manuscript are included in the main text and supplementary information and will also be available from the corresponding authors upon reasonable request.

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

## Acknowledgements

We are grateful to the Coulter Translation Program at the University of Missouri and the National Institutes of Health for support of this work through HG009338 (L.-Q.G.), GM114204 (L.-Q.G.) and GM063732 (S.-J.C.).

## Author contributions

X.Z., K.T. and L.-Q.G. designed the experiments. X.Z., K.T., R.S., X.D., A.J.B. and J.W. performed the experiments and analyzed the data. D.Z., C.Z. and S.-J.C. designed the simulations. D.Z. and C.Z. performed all the simulations. S.-J.C. and L.-Q.G. built the pathway model. X.Z., D.Z., S.-J.C. and L.-Q.G. wrote the paper. All authors contributed to the manuscript revision.

## Additional information

**Competing interests:** The authors declare no competing financial interests.

