## [Peer Review File · Nature Communications]

Reviewers' comments:

Reviewer #1 (Remarks to the Author):

In the article "Nanopore electric snapshots of RNA tertiary folding pathway" the authors propose a new application of the nanopore technology to study the RNA folding pathway. In general, this is a very interesting article, however I have a number of critical comments, questions, and suggestions for improvement:

Major remarks:

I am not entirely convinced that the data presented agree with the proposed model. The agreement between the fitting of the number of structural subpopulations with the histograms (Figure 2, S1 and S2) is rather poor. On the other hand, the fitting on Figure 5 the fitting is convincing, hence in general a better fit on Figure 2, S1, and S2 could be obtained? This is a very fundamental issue for the whole story presented in this article.

As for histograms, it is not clear what is presented on the y-axis. If N% is the percentage of nanopores with a particular characteristics, then what are the data for the rest (majority) of the pores? (the height of the histogram bars rarely reaches 0.2).

Fig. 4a - in my opinion, the curve for the PK structure clearly shows that even after 30 seconds it has not reached equilibrium, while the equilibrium state is quite fundamental to reach any conclusions.

Overall, the article presents only one "proof of concept" application, but unfortunately does not discuss potential practical applications.

Limitations of the method are also not discussed. For instance, can this approach be applied to RNA molecules composed of multiple chains? RNA molecules with modified residues? RNA molecules that are very small? RNA molecules that contain mostly non-canonical base pairs (and very little canonical secondary structure)? What about size limitations (e.g., would it be possible to study the folding of ribosomal RNA?)

It is completely unclear how the experimental data is integrated with the computational folding analysis. It seems to me that exactly the same computational results could be obtained without the experimental data, by performing folding simulations for RNA fragments with increasing length. So is the RNA folding used only to provide 3D illustrations for the experimental data, or does it contribute to the actual result?

Finally, to which extent the results depend on the use of the Vfold method? Would the results change if another RNA 3D structure modeling/folding method was used? Please comment on the use of other methods, e.g. regular molecular dynamics, RNAComposer, other coarse-grained methods SimRNA, FARNA, etc.

Minor remarks:

Introduction:

"Small angel" should be "small angle"

In Fig. 5, panels a, b and d similar colors are used to delineate key aspects of the drawing. Unfortunately, a given color means something completely different in each of these panels, which is extremely confusing. Perhaps the coloring could be unified?

p 12:

"The folding equilibrium within an RNA pseudoknot is highly dependent on RNA sequence .." should be "The folding equilibrium within an RNA pseudoknot is highly dependent on RNA sequence .."

p 13:

"In combination with simulations, this system will first disrupt any secondary or tertiary structures the biomolecule has and let it start to fold from totally disrupted structure". This sentence suggests that simulations somehow influence the experimental system, which is obviously not the case.

Reviewer #2 (Remarks to the Author):

This manuscript presents an interesting study combining nanopore, single molecule measurements with simulation to provide insight into states that can be formed by the single stranded molecule with the sequence of the T2 gene 32 pseudoknot, and that can, eventually fold to the pseudoknot.

The identification of the states, while interesting, may not be terribly new, similar states were identified in a previously published account by the same authors (Ref 44); however that work focused on unfolding. What I like about this manuscript is the use of simulation to create a master equation, connecting the identified states to create an RNA folding pathway. Their arguments are convincing and compelling.

I have a number of minor comments and questions that can help improve the manuscript:

First of all, seems like more information could be extracted by the change in the conductance. For example, is there any way to determine the length of RNA structure unfolded? The figures suggest that the pore has a significant length, and can be populated by both RNA and the DNA linker. This would be a valuable validation of the model, for example, if the change in conductance could be explained by a channel that contained 60% RNA and 40% DNA. Thus the unfolded length would serve as a check on (partially folded) structures.

Second, for the first part of the manuscript, the impression is given that the molecule folds within 10 seconds (the data shown in Figure 2 represent a PK that folds in 10 s). Thus it took several reads for me to understand that the 'folding time' is a variable in the problem, and that it would be changed. The language, as written, suggests that folding is complete within 10 seconds, therefore the identification of alternative states was quite confusing. Once I reached the section describing folding kinetics, and the variation of the folding time, it became clear, but I believe it would clarify the manuscript to introduce the concept of kinetic experiments at an earlier time.

Is the ionic strength of the solution (- Mg) reported in the body of the paper? Please add this.

The supporting information was very thorough, however, could a reference be provided for equation S1?

The manuscript would benefit from a careful proof reading.

Finally Figure 3 obscures the important observation of 'two levels' in the conductance curves. Please add an inset that clearly shows the levels.

Reviewer #3 (Remarks to the Author):

The manuscript by Zhang et al reports on a single-molecule kinetic assay for probing RNA folding in a well-known RNA pseudoknot sequence. The manuscript is well-written, analysis is consistent and logical, and the results are sound. Apart from several minor comments, I believe the manuscript is suitable for publication in Nature Comm., because of the neat way the authors used to elucidate so many intermediates in the RNA structure based on their voltage-induced unfolding times.

- 1) In the introduction, the authors missed a couple of key works involving nanopores and folded RNA structures: Shasha et al. (ACS Nano, 2014, p-6425) detected RNA conformational changes upon drug binding, and Henley et al. (Nano Letters, 2016, p-138) used machine learning algorithms to distinguish tRNAs.
- 2) In Figure 2a, the label "tau_unfolding time" goes all the way past the molecular escape (labeled as G), shouldn't unfolding be until G, but not including G?
- 3) The transient times associated with membrane RC time is quite large (~5 ms), the authors may want to consider improving their setup configuration to allow probing of even faster intermediates in future studies.

Reviewer #1 (Remarks to the Author):

In the article "Nanopore electric snapshots of RNA tertiary folding pathway" the authors propose a new application of the nanopore technology to study the RNA folding pathway. In general, this is a very interesting article, however I have a number of critical comments, questions, and suggestions for improvement:

Major remarks:

I am not entirely convinced that the data presented agree with the proposed model. The agreement between the fitting of the number of structural subpopulations with the histograms (Figure 2, S1 and S2) is rather poor. On the other hand, the fitting on Figure 5 the fitting is convincing, hence in general a better fit on Figure 2, S1, and S2 could be obtained? This is a very fundamental issue for the whole story presented in this article.

According to the reviewer's suggestion, we intensively repeated all relevant experiments to improve the fitting and more clearly identify different time components in histograms of these figures. Through these experiments, we collected a greatly increased numbers of single-molecule events, which are required for more convincing identification and better fitting of structural subpopulations.

In addition to Figure 3, S1 and S2 that have been suggested by the reviewer, we also improved Figure S3, including histograms and fittings at all folding times from 1 s to 60 s. Note that the histogram and fitting at the 60 s folding time in Figure S3 is a new experiment.

The total numbers of events (N_0) collected for histograms before and after improvement are summarized in the following table. N_0 values are provided in the revised manuscript.

	N_0 (Before improvement)	N_0 (After improvement)	Total
Figure 3	263	644	907
Figure S1	388	756	1144
Figure S2,	178	383	561
Figure S3 (1 s)	113	548	661
Figure S3 (2 s)	163	568	731
Figure S3 (5 s)	187	476	663
Figure S3 (10 s)	263	578	841
Figure S3 (30 s)	182	644	846
Figure S3 (60 s, new)		567	567

As for histograms, it is not clear what is presented on the y-axis. If N% is the percentage of nanopores with a particular characteristics, then what are the data for the rest (majority) of the pores? (the height of the histogram bars rarely reaches 0.2).

In the revised manuscript, we changed the y-axis label from "N%" to " N/N_0 ". N/N_0 is more meaningful, in which N_0 is the total number of events used to construct the histogram, and N is the number of events within each log-scale bin time along the x-axis. Thus N/N_0 represents the

fractional population within each log bin time. Importantly, this method of data presentation allows the sum of N/N_0 for all bins, i.e. the area of the histogram, to be normalized to “1”. After fitting, the area covered by each identified time component (P) is equal to the fractional population of the corresponding folding structure. P values for each structure at different folding times, i.e. populational kinetics ($P-t_{fold}$ curve) were plotted in Figure 4a.

In the revised manuscript, N/N_0 was described in Figure 3 legend.

The unfolding durations for different folding structures vary dramatically in 3-4 orders of magnitude from milliseconds to seconds. To accurately determine all time components in a histogram, we used log time in the x-axis, i.e. the unfolding duration was binned in the log scale. To clearly demonstrate the time in log scale, in revised histograms of Figure 3, S1, S2 and S3, we changed the tick labels to “1”, “10”, “100”, “1000”, and “10,000”, and changed the x-axis title to τ (ms).

Fig. 4a - in my opinion, the curve for the PK structure clearly shows that even after 30 seconds it has not reached equilibrium, while the equilibrium state is quite fundamental to reach any conclusions.

We agree that the equilibrium state is important in the RNA folding pathway. To detect the equilibrium state, we extend the folding time to $t_{fold}=60$ s, and investigated the population distribution of different folded structures with this extended folding time. The histogram of unfolding duration and the population fitting for the four components (including intermediates and pseudoknot) with 60 s folding time are added in revised Figure S3. The population kinetics ($P-t_{fold}$ curve) for all components (obtained from the histograms) in revised Figure 4a are now extended to 60 s folding time. In addition, standard deviation for each data point is provided. The transition rates between different intermediated states shown in Figure 4b have been slightly modified based on the new experimental data.

The figure below is similar to Figure 4a, but the fitting curves have been extended to $t_{fold}=100$ s. If the equilibrium population is P_0 , the deviation of the population at $t_{fold}=60$ s to the equilibrium populations should be $(P_{t_{fold}=60s}-P_0)/P_0$. This deviation for each state is shown in the parentheses at $t_{fold}=60$ s. These deviations are small, slightly varying between 1.7%–3.6%, supporting that the folding reaches equilibrium at $t_{fold}=60$ s.

Overall, the article presents only one "proof of concept" application, but unfortunately does not discuss potential practical applications. Limitations of the method are also not discussed. For instance, can this approach be applied to RNA molecules composed of multiple chains? RNA molecules with modified residues? RNA molecules that are very small? RNA molecules that contain mostly non-canonical base

pairs (and very little canonical secondary structure)? What about size limitations (e.g., would it be possible to study the folding of ribosomal RNA?)

Thanks the reviewer for the good suggestion. We analyzed the limitations, improvements and broader applications of this nanopore-based RNA folding approach in the Discussion section (Page 14-16). Specifically,

Limitations and improvement

To identify more intermediate states, such as the pre-pseudoknot structure with two native helices formed but without loop-helix tertiary interactions, and to develop further applications, the resolution and accuracy of the nanopore system needs improvement. Firstly, due to the resistor-capacitor (RC) time of the setup, the voltage change will simultaneously cause a curved charging current (Fig. 2b and 3b). Fast unfolding events occurring in this sharply changing current region could be unidentifiable. To improve the temporal resolution for fast events detection, the RC time of the setup needs optimization. Meanwhile application of low voltage is an effective strategy to significantly prolong the unfolding duration for identifying fast partially-folded structures. Secondly, it could be difficult to discriminate different folding structures that are similar in unfolding duration (Fig. 3a), due to their large overlap in the duration histogram. To solve this problem, we can use the blocking level as an “identifier” to discriminate different folding structures. This hypothesis was verified to be feasible through our new experiments. As shown in Fig. S4, different RNA lengths and positions in a DNA/RNA chimera can be clearly discriminated based on their blocking levels in the nanopore, offering the potential for using the blocking level to precisely identify RNA folding states. Thirdly, our method may be suitable to long RNAs investigation, but if limited, alternative methods including nanopore sequencing could be used to precisely read RNA positions and report various folded states. Notably, synthetic nanopores with tunable dimension have been developed recently to characterize drug-induced RNA conformational changes. Combined with machine learning, synthetic nanopore can discriminate different tRNAs.

From the theoretical aspect, a more physically reliable force field in the MD simulation is helpful to provide structural details for various potential folding intermediate states and then to understand the nanopore unfolding current signature. The key points include non-canonical base pairs interaction, multiple base-base interaction, nucleotide-ion interaction, free of native-biased and ability to account for the whole folding landscape, and more efficient conformational space sampling. Furthermore, a direct simulation of the nanopore unfolding experiment by, such as Steered Molecular Dynamics (SMD) and nanopore translocation simulation, may be very helpful to understand the unfolding electric current signatures of various structures and to establish the direct connection between electric current signature and RNA structure. Improved theoretical analysis would significantly improve the resolution of nanopore technique, making it possible to more accurately reveal RNA intermediate states and folding pathways and broadening the nanopore applications in biomolecular folding study.

Broad applications

The improved system could be expanded to study a variety of disease-relevant RNA and DNA tertiary structures, from hairpins and pseudoknots to kissing-loop and G-quadruplexes, from short microRNA-target interaction to ribosome RNA. It can also be applied to study the interaction of ligands with RNA targets such as RNA repeat and riboswitches. In addition to mutant construction, modified nucleotides can be introduced to detect chemical-specific folding procedure in the nanopore. Therefore, the nanopore can be potentially combined with chemical approaches such as Selective 2'-Hydroxyl Acylation analyzed by Primer Extension (SHAPE) for joint structure exploration.

Most notably, our nanopore system can provide very small forces (by applying low voltage) to detect weak interactions in small RNA structures; and as demonstrated by this and previous studies, this system is powerful to characterize non-canonical base pairs involved in the tertiary structure formation and dissociation. Finally, this system may be generalized to investigate the folding of other biomolecules, including polymers, peptides, and even proteins. Overall, this method can find applications in biomolecule folding investigation and molecular folding related areas such as pharmaceutical kinetic investigation and drug development.

It is completely unclear how the experimental data is integrated with the computational folding analysis. It seems to me that exactly the same computational results could be obtained without the experimental data, by performing folding simulations for RNA fragments with increasing length. So is the RNA folding used only to provide 3D illustrations for the experimental data, or does it contribute to the actual result?

Thanks the reviewer for raising this important question and we are sorry for the confusion. Overall, the nanopore unfolding experiment provided the current signatures and global structure features for all intermediated states (including the final pseudoknot) and the time-dependent folding population for each concerned state, the coarse-grained (CG) Molecular Dynamics (MD) simulation was employed to search the potential kinetics trapped (intermediated) states and the corresponding structural details and then to help understand the nanopore unfolding current signature. The master equation approach was used to explore the time-dependent folding populations and to extract the transition rates between different intermediated states, and the combination of those three methods result in the construction of folding pathway.

In detail, the nanopore unfolding experiments provided the current signatures, including the number of electric current block levels and the unfolding time scale, for the identified intermediated states. Based on two controlled experiments (one is the unfolding of T2 pseudoknot in the previous study (Zhang et al, JACS), the other is the unfolding of the ref-HP here), those current signatures can be used to determine the global features, such as pseudoknot-like (two-level block pattern) or stem-loop (one-level block pattern) structure (Fig. 3b), and the relative stability (in general, for the same type of intermediated structures, the stability is increased with increasing the number of base pairs) for each intermediated state, but the detailed structures information is in absence. Only the combination of nanopore unfolding experiment and CG MD simulation can reconstruct the structural details for each intermediated state. In the CG MD simulation, three main kinetics trapped states (two stem-loop structures, one pseudoknot-like) were observed before the final pseudoknot was formed and this is in well agreement with the experiment results. The structure information for the intermediate states indicated by CG MD simulation is helpful to understand the nanopore unfolding current signatures and then to establish the connection between the intermediates states identified by the nanopore and observed in simulation. However, as the REMD (replica-exchange MD) enhanced sampling technique was employed to rapidly search the conformation space, the time-dependent folding population for each intermediate state and the related transition rate between different intermediate states are not captured in CG MD simulation. Therefore, the master equation approach was used to extract the transition rates between different intermediated states from the experimentally determined time-dependence of the populations. In conclusion, the integration of nanopore-based experimental data with computational folding analysis can provide information about the folding pathway shown in Figure 4b (main text).

In general, due to the limitations in force field, such as the non-canonical base-base interaction and nucleotide-ion interaction, and insufficient sampling in conformation space, the folding results obtained by computational simulation sometimes could be less reliable and the confirmation of

computational results by experimental data is essential. For instance, not only the on-pathway intermediated states transition (such as SS→HP1→PK), but also the off-pathway misfolded states (SS→HP2, HP1→TS) were observed here, and this point was cross validated by both experimental data and computational simulation. Additionally, the experimental data also provide the information about time-dependent folding populations for all the intermediated and the related transition rates, which are also very important for the study of folding pathway. Thus, the RNA folding analysis presented here was used to provide 2D and 3D illustrations to understand the experimental data and it partly contribute to the actual result.

To further clarify this issue, we have also added additional explanations in the main text; See the last paragraph on page 13 and the first two lines on page 14.

Finally, to which extent the results depend on the use of the Vfold method? Would the results change if another RNA 3D structure modeling/folding method was used? Please comment on the use of other methods, e.g. regular molecular dynamics, RNAComposer, other coarse-grained methods SimRNA, FARNA, etc.

Thanks the reviewer for the helpful suggestion. As mentioned above, the kinetics intermediate states observed in the folding simulations are consistent with the signals detected in the nanopore unfolding experiment. In fact, the misfolded HP2 and TS states identified by the CG MD simulation are also supported by other computational models. Following the reviewer's suggestion, we have used other folding models to run the calculations. The folding simulations of T2 pseudoknot by SimRNA model also found a misfolded stem-loop structure (2D structure: ((((((.(.(((.....))))).))))..)), Bujnicki et al, Nucleic Acid Res 44, e63), which is very similar to the HP2 structure obtained here (2D structure: ((((((.(.(((.....))))..))))..)). And the Vfold-based 2D structure free energies analysis suggested that the TS misfolded state (free energy: -9.8 kcal/mol) may be treated as a misfolded suboptimal state.

In the 3D structure level, depending on the algorithms of the different 3D structure modeling methods, the 3D structures predicted by Vfold, RNAComposer, SimRNA and FARNA are slightly different, as shown in Table 2 below. For the native PK structure, the CG MD model used in this study provides the best prediction (all-heavy atom RMSD: 2.88 Å) among all the above modeling methods. For the other four intermediate states (HP1 and HP2 for the T2 RNA folding, HP3 and PK* for the poly-U loop mutation folding), the Vfold3D, RNAComposer and SimRNA provide similar global 3D structures with RMSDs all less than 5 Å, while the FARNA failed to predict the correct global fold for the pseudoknot-like structures (PK*, PK, and TS). For the TS state, due to the lack of templates in the database (PDB dataset), the template-based methods such as Vfold3D and RNAComposer may give inconsistent results. For the SimRNA, the base pairs and helices of the TS state (which is confirmed both by different computational models and the experimental data) are altered in the 3D structure prediction process. Overall, the use of other RNA 3D structure modeling methods only slightly changes the results here.

The table below shows all-heavy atoms RMSD values (Å) of the 3D structures for all the intermediated states as predicted by the different programs. For the final pseudoknot PK, the RMSD values (shown in the row "PK") are calculated with respect to the native structure (PDB id: 2tpk). For other intermediated states, the RMSD values are calculated with respect to the CG MD-predicted structures used in the study. HP1, HP2, TS and PK are the intermediated states for T2 RNA folding, HP3 and PK* are the intermediated states for the folding of the poly-U loop mutation. The 5'- and 3'-tails in HP1, HP2 and HP3 are not included in the calculation of RMSD values. The 2D

structures identified by the CG MD model are used as the input for the modeling methods and the first structure/cluster predicted by each program is chosen as the predicted structure for comparisons. The predictions were obtained from

Vfold3D (<http://rna.physics.missouri.edu/vfold3D2/>),
RNAComposer (<http://rnacomposer.cs.put.poznan.pl/>, batch mode),
SimRNA (<http://genesilico.pl/SimRNAweb/submit>) and
FARNA (modeling by Rosetta, following the guide at https://www.rosettacommons.org/demos/latest/public/rna_assembly/README).

	CG MD (this study)	Vfold3D2	RNAComposer	SimRNA	FARNA
HP1	0	3.29	3.06	1.81	2.78
HP2	0	4.73	4.30	3.96	3.47
TS	0	6.99	9.30	7.09	18.42
PK	2.88	3.26	4.86	3.67	20.74
HP3	0	1.74	4.12	2.94	2.60
PK*	0	3.22	4.42	3.12	19.98

Minor remarks:

Introduction: "Small angel" should be "small angle".

This typing mistake was corrected in revised Introduction, line 6 Page 3

In Fig. 5, panels a, b and d similar colors are used to delineate key aspects of the drawing. Unfortunately, a given color means something completely different in each of these panels, which is extremely confusing. Perhaps the coloring could be unified?

We accepted the reviewer's suggestion and revised Figure 5. We changed panel d to the grey-level plot. We did not change panel a, which uses colors to mark different fragments of T2 mutant RNA, and panel b, which uses color to separate observed structures of T2 mutant RNA. The reason for only changing Panel d is that Panel d is independent to panels a and b. It comprehensively compares the unfolding durations of observed folding structures between the WT and mutant RNAs, and between the presence and absence of Magnesium ions. The grey-level illustration is simple and clear enough to separate different conditions.

p 12: "The folding equilibrium within an RNA pseudoknot is highly depend on RNA sequence .." should be "The folding equilibrium within an RNA pseudoknot is highly dependent on RNA sequence

This typing mistake was corrected in the revised manuscript. Please see Line 12 on Page 14.

p 13: "In combination with simulations, this system will first disrupt any secondary or tertiary structures the biomolecule has and let it start to fold from totally disrupted structure". This sentence suggests that simulations somehow influence the experimental system, which is obviously not the case.

We agree with the reviewer, and changed the claim to

"The nanopore system in this report provides a novel tool for investigating RNA folding process. The nanopore first disrupts any secondary or tertiary structures of the target RNA molecules, and then let RNA start to fold from the unfolded state." See Line 5 on Page 13.

Reviewer #2 (Remarks to the Author):

This manuscript presents an interesting study combining nanopore, single molecule measurements with simulation to provide insight into states that can be formed by the single stranded molecule with the sequence of the T2 gene 32 pseudoknot, and that can, eventually fold to the pseudoknot.

The identification of the states, while interesting, may not be terribly new, similar states were identified in a previously published account by the same authors (Ref 44); however that work focused on unfolding. What I like about this manuscript is the use of simulation to create a master equation, connecting the identified states to create an RNA folding pathway. Their arguments are convincing and compelling.

I have a number of minor comments and questions that can help improve the manuscript:

First of all, seems like more information could be extracted by the change in the conductance. For example, is there any way to determine the length of RNA structure unfolded? The figures suggest that the pore has a significant length, and can be populated by both RNA and the DNA linker. This would be a valuable validation of the model, for example, if the change in conductance could be explained by a channel that contained 60% RNA and 40% DNA. Thus the unfolded length would serve as a check on (partially folded) structures.

As shown in Figure 3, different RNA folding structures can be discriminated according to their unfolding duration in the nanopore, which dramatically varies from 1 ms (e.g. short hairpin) to 1,000 ms scales (e.g. pseudoknot). In this question, the reviewer asked whether the blocking level (blockade current amplitude) can be used as an identifier to discriminate partially folded structures, supposing that trapping RNA to different depths in the nanopore results in different blocking levels. In the revised manuscript, we experimentally verified this possibility. See new Figure S4 in Supplementary Information. The result was also discussed in Discussion, Line 3-7 on Page 15.

Specifically, we designed a series of DNA/RNA chimeras. RNAs in these chimeras have different lengths and locations. When immobilized in the nanopore by the streptavidin attached, these RNA fragments will occupy the α HL pore at various positions. As shown in Figure S4 (the panel b and c are shown below), we found that each chimera generates a specific blocking level that distinguishes itself from other chimeras. This finding suggests that the length and position of RNAs in the nanopore can be discriminated based on the characteristic blocking levels. These blocking levels are determined by the sequence of DNA and RNA. Therefore, the blocking level potentially can be used as an identifier to report partially folded RNA structure.

For T2 RNA in this report, we have verified that the blocking patterns can be used to discriminate certain folding structures, e.g. two-level blockades for the TS and PK states and single-level blockades for HP1 and HP2 states. However, it is still difficult to read all T2 RNA folding structures from the blocking levels. One reason is that the unfolding of these structures occurs within the highly curved charging current region (RC time) when the voltage jumps to -60 mV (Figure 3b). The current amplitude of the blockade within this region is difficult to measure. Another reason is that, as shown in Figure 3 and 4, these folding structures do not contain or contain a very short (2-3 nucleotides) overhang RNA (3' end), therefore they may not significantly influence the nanopore conductance when trapped in the pore. However, in the future study, we can consider to extend the RNA overhang to report the position of dissociated RNA fragment in the nanopore.

Second, for the first part of the manuscript, the impression is given that the molecule folds within 10 seconds (the data shown in Figure 2 represent a PK that folds in 10 s). Thus it took several reads for me to understand that the 'folding time' is a variable in the problem, and that it would be changed. The language, as written, suggests that folding is complete within 10 seconds, therefore the identification of alternative states was quite confusing. Once I reached the section describing folding kinetics, and the variation of the folding time, it became clear, but I believe it would clarify the manuscript to introduce the concept of kinetic experiments at an earlier time.

As the reviewer suggested, we added a paragraph in the middle of Page 6 briefly introducing the concept of the method prior to specific kinetic experiments.

"The overall strategy for the nanopore folding study is as follows: A positive voltage is firstly applied to release an unstructured probe molecule into the trans solution; This unstructured molecule is held in the trans solution for a pre-defined duration, i.e. folding time (such as 1 s, 10 s or 30 s) for re-folding; At the end of the folding time, a negative voltage is applied to pull the folded RNA back to the cis solution. From the RNA unfolding signature, its folding structure (formed during the folding time) is inferred. By repeating this protocol, a large number of single-molecule snapshots are measured, classified, and assigned to specific folding structures with a fractional population. By

varying the folding time, time-dependent folding populations can be obtained to elucidate the folding pathways.”.

Is the ionic strength of the solution (- Mg) reported in the body of the paper? Please add this.

The RNA folding experiment in the presence Mg^{2+} was done in 1 M NaCl and 10 mM $MgCl_2$. The experiment without Mg^{2+} was done in 1 M NaCl without $MgCl_2$. Both ionic strength conditions were described in Page 6 and Page 12, and described in detail (Supplementary Information S1).

The supporting information was very thorough, however, could a reference be provided for equation S1?

The log-binned histogram is in particular suitable to separating and quantifying time components that are largely different to each other. The method of analyzing log-binned histograms were firstly proposed in Blatz & Magleby *J. Physiol.* 378, 141-174 (1986), and theoretically developed in Sigworth & Sine, 52, 1047-1054 (1987). Both references are cited in the revised Supplementary Information for Equation S1.

The manuscript would benefit from a careful proof reading.

The manuscript has been proofread by colleagues.

Finally Figure 3 obscures the important observation of ‘two levels’ in the conductance curves. Please add an inset that clearly shows the levels.

This is a good suggestion. We added two insets in Figure 3b to clearly show the two-level conductance pattern for both the TS and PK states, which are distinct from the single-level conductance pattern for other intermediate states SS, HP1, and HP3.

Reviewer #3 (Remarks to the Author):

The manuscript by Zhang et al reports on a single-molecule kinetic assay for probing RNA folding in a well-known RNA pseudoknot sequence. The manuscript is well-written, analysis is consistent and logical, and the results are sound. Apart from several minor comments, I believe the manuscript is suitable for publication in Nature Comm., because of the neat way the authors used to elucidate so many intermediates in the RNA structure based on their voltage-induced unfolding times.

1) In the introduction, the authors missed a couple of key works involving nanopores and folded RNA structures: Shasha et al. (ACS Nano, 2014, p-6425) detected RNA conformational changes upon drug binding, and Henley et al. (Nano Letters, 2016, p-138) used machine learning algorithms to distinguish tRNAs.

Thank the review for introducing two recent works on nanopore RNA detection. Both papers report the design of synthetic solid nanopores for RNA detection. Due to scalable pore size, synthetic nanopore has been demonstrated in these works to be powerful to discriminate various tRNAs and RNAs that change structures upon drug binding. We believe that the advantages of protein pore and synthetic pore allows their combined use to comprehensively investigate RNA structure and biological mechanism.

2) In Figure 2a, the label “tau_unfolding time” goes all the way past the molecular escape (labeled as G), shouldn't unfolding be until G, but not including G?

The reviewer is correct. The unfolding duration should not include the G state in Figure 2, as the unfolded polymer in this state has been pulled out of the pore. It was corrected in revised Figure 2.

3) The transient times associated with membrane RC time is quite large (~5 ms), the authors may want to consider improving their setup configuration to allow probing of even faster intermediates in future studies.

The curved charging current within the membrane RC time is an issue. Reducing the RC time would increase the resolution for probing faster intermediate structures. The setup needs improvement to shorten the RC time. For example, decreasing the setup capacitance, such as using miniaturized nanopore device and formation of smaller membrane patch, could efficiently shorten the RC time. Another approach to increasing the fast event detection capability is prolonging the unfolding duration of target molecules. For example, the same folding structure can take longer duration to unfold when a small voltage that provides a small pulling force is used. This approach has been verified in various previous reports on nanopore DNA unzipping and in the current study at -60 mV (Figure 2) and -120 mV (Figure S1).

REVIEWERS' COMMENTS:

Reviewer #1 (Remarks to the Author):

The Authors have addressed all my suggestions, and performed the requested analysis. I am fully satisfied.

I only have one minor remark: Many references in the literature list are misformatted. Some journal names are shown in different formats (e.g., RNA and Rna), and in many papers page and/or issue numbers are missing, e.g. in references 8, 29, 40, 60 (actually, this article is from 2016, not 2015), 64, 76.

Reviewer #2 (Remarks to the Author):

The authors have fully addressed the concerns I raised in the initial review. I recommend publication.

Reviewer #3 (Remarks to the Author):

I believe the authors have addressed all of the reviewers' comments satisfactorily, including the statistical concerns. One standing issue is how the applications for unknown RNA structures will be handled, particularly if timescales for various processes are degenerate (i.e., a fast and a slow process may overlap with just a slow process).

Otherwise, I believe the manuscript is suitable for publication in this state.

Reviewer #1 (Remarks to the Author):

The Authors have addressed all my suggestions, and performed the requested analysis. I am fully satisfied.

We highly appreciate the reviewer's evaluation. The reviewer's important suggestions have greatly promoted the significance and impact of this report, enabling us to present an accurate frontier research work to broader readers.

I only have one minor remark: Many references in the literature list are misformatted. Some journal names are shown in different formats (e.g., RNA and Rna), and in many papers page and/or issue numbers are missing, e.g. in references 8, 29, 40, 60 (actually, this article is from 2016, not 2015), 64, 76.

According to both the reviewer and editor's suggestion, we have reformatted all the references based on the journal style. References have been screened to ensure that their publishing information is correct. In addition, as required, the total number of references has been controlled to be within 70.

Reviewer #2 (Remarks to the Author):

The authors have fully addressed the concerns I raised in the initial review. I recommend publication.

We highly appreciate the reviewer's evaluation. The reviewer's important suggestions have greatly promoted the significance and impact of this report, enabling us to present an accurate frontier research work to broader readers.

Reviewer #3 (Remarks to the Author):

I believe the authors have addressed all of the reviewers' comments satisfactorily, including the statistical concerns. One standing issue is how the applications for unknown RNA structures will be handled, particularly if timescales for various processes are degenerate (i.e., a fast and a slow process may overlap with just a slow process).

We thank the reviewer's insightful question. We suggest that the nanopore snapshot approach needs to be combined with theoretical analysis such as simulation, to elucidate unknown RNA structures. As shown in the paper, the nanopore is powerful to accurately identify various folding intermediate states based on their stabilities and their nanopore conductance signatures, but needs to be correlated with the simulation results to assign each observed signatures to a specific structure. The specific analysis is described in page 13 of Discussion section and shown below.

"The nanopore snapshot-based kinetic detection, combined with CG molecular dynamics-based structural modeling, master equation-based rate constant estimation, and population kinetics analysis, provides a tool to investigate the RNA folding process. Through a programmable RNA disruption-refolding-disruption procedure in the nanopore, a series of intermediates that may be inaccessible in equilibrium experiments can be captured based on their unfolding signatures and the population kinetics for each state can be measured. The nanopore is able to capture a wide range of intermediates, with the unfolding duration ranging from milliseconds to seconds and minutes. The CG MD simulations can identify the potential intermediate structures and kinetic pathways, which are supported by the nanopore signatures. Based on the identified structures, the master equation approach can provide the transition rates between different states from the population kinetics. The combination of the above three methods leads to a reliable construction of the folding kinetics. The

reliability of the results is supported by the theory-experiment consistency for the different measured properties. The integrated approach above can potentially be adapted for the study of RNAs with unknown structures. For RNAs with unknown structures, in addition to the structure information provided from the computational studies, nanopore data for various designed mutants would be highly useful for the probing and confirmation of the structures, stabilities, and folding pathways. The theory-experiment comparisons for the designed mutants may lead to reliable identification of the key factors that determine the kinetics.”

Otherwise, I believe the manuscript is suitable for publication in this state.

We highly appreciate the reviewer’s evaluation. The reviewer’s important suggestions have greatly promoted the significance and impact of this report, enabling us to present an accurate frontier research work to broader readers.